

# Computer-aided colorectal cancer diagnosis: AI-driven image segmentation and classification

Çağatay Berke Erdaş

Computer Engineering, Başkent University, Ankara, Turkey

## ABSTRACT

Colorectal cancer is an enormous health concern since it is among the most lethal types of malignancy. The manual examination has its limitations, including subjectivity and data overload. To overcome these challenges, computer-aided diagnostic systems focusing on image segmentation and abnormality classification have been developed. This study presents a two-stage approach for the automatic detection of five types of colorectal abnormalities in addition to a control group: polyp, low-grade intraepithelial neoplasia, high-grade intraepithelial neoplasia, serrated adenoma, adenocarcinoma. In the first stage, UNet3+ was used for image segmentation to locate the anomalies, while in the second stage, the Cross-Attention Multi-Scale Vision Transformer deep learning model was used to predict the type of anomaly after highlighting the anomaly on the raw images. In anomaly segmentation, UNet3+ achieved values of 0.9872, 0.9422, 0.9832, and 0.9560 for Dice Coefficient, Jaccard Index, Sensitivity, Specificity respectively. In anomaly detection, the Cross-Attention Multi-Scale Vision Transformer model attained a classification performance of 0.9340, 0.9037, 0.9446, 0.8723, 0.9102, 0.9849 for accuracy, F1 score, precision, recall, Matthews correlation coefficient, and specificity, respectively. The proposed approach proves its capacity to alleviate the overwhelm of pathologists and enhance the accuracy of colorectal cancer diagnosis by achieving high performance in both the identification of anomalies and the segmentation of regions.

## INTRODUCTION

Colorectal cancer, or colon cancer, is a highly lethal form of cancer. Moreover, it is the third cancer type with the highest incidence in women and the fourth cancer type with the highest occurrence in men worldwide. In addition, colorectal cancer accounts for 10% of total cancer instances (*Sung et al., 2021*). Histopathological examination is an essential component of diagnosing and treating colorectal cancer. It is considered the most reliable diagnostic tool for assessing the intestinal canal and is crucial for effective treatment (*Pamudurthy, Lodhia & Konda, 2019*). The advantage of intestinal tissue sampling by intestinal biopsy is that it helps to ascertain the actual health condition of the patient, although it causes minimal harm to the physical condition and tends to heal quickly. Following that, the biopsy tissue is divided into sections and colored using haematoxylin &

Corresponding author
Çağatay Berke Erdaş,
berkeerdas@gmail.com

eosin (H&E). H&E coloring is one of the most widely used technique for highlighting tissue characteristics and revealing details between the nucleus and cytoplasm in tissue slices (*Labianca et al., 2013*; *Fischer et al., 2008*). When examining the colon, the pathologist first examines the histopathological sections for conformity and locates the lesion. A low-magnification microscope is thereafter employed to analyze and diagnose the pathology sections. Nevertheless, the following issues are often encountered during the process of diagnosis: The microscope can be set to high magnification when fine structures need to be observed in more detail. At higher magnifications, the depth of field may become increasingly shallow, which can result in a narrower focal plane. This may make it challenging to maintain sharpness across the entire sample, especially when examining complex details that extend beyond the limited focal range. However, problems such as the variability of diagnostic results depending on the subjectivity of different doctors, doctors may miss some important information when there is a vast quantity of test data, and analyzing a significant amount of previously gathered data is challenging (*Chan, 2014*). Therefore, these issues need to be addressed effectively.

The utilization of computer-aided diagnosis (CAD) has facilitated precise and efficient pathological examination of each case individually through computers (*Gupta et al., 2021*). One of the main features of CAD includes image segmentation, which pathologists can use as important evidence in their diagnostic process. Furthermore, the classification of the type of anomaly in the histopathological image is of great importance. The type of anomaly involved can provide important information about the individual's risk of developing cancer. With the developing technology, segmentation can be used to identify the type of segmented object. However, in the case of histopathological images, this application may not be directly applicable. This is because histopathological images typically contain only one type of object, which is a specific tissue or anomaly. In other words, pathologists can only apply segmentation for the suspected type of anomaly. This problem is the biggest obstacle to the fully automatic operation of CAD-supported systems. To overcome this problem, using CAD, anomalies in the histopathological image can be highlighted by segmentation and the type of these anomalies can be detected using deep learning models. Thus, a fast, effective objective, fully automatic decision support system can be developed to reduce the workload of pathologists.

This study focuses on the segmentation and classification of histopathological colorectal cancer sections including five tumor anomaly types: polyp, low-grade intraepithelial neoplasia, high-grade intraepithelial neoplasia, serrated adenoma and adenocarcinoma, and a control group. In the first stage, all histopathological images of different types were segmented with UNet3+, and the resulting mask and raw images were multiplied, leaving only the anomalous parts in the images, in other words, highlighting the anomalies. These highlighted images were classified with the Cross-Attention Multi-Scale Vision Transformer (CrossVIT) deep learning model and the anomaly type was determined. In this way, anomalies in histopathology images were automatically detected without the need for any intervention and/or prior knowledge.

## RELATED STUDIES

Related studies in the literature can be divided into two categories: segmentation and classification. *Martos et al. (2023)* addressed the problem of optimization of the segmentation and detection of the nuclei in images of gastric cancer. They used blot normalization and blur aberration removal methods to enhance the accuracy of this process. In their research, the authors evaluated and compared different digital image analysis techniques for gastric cancer. They proposed a color normalization and spot separation method that uses adaptive local thresholding to segment nuclei, and a marker-controlled watershed segmentation. As a result of the study, F1-measure and adaptive Jaccard Index scores of $0.854 \pm 0.068$ and $0.458 \pm 0.097$ were obtained. *Gao et al. (2023)* focused on a complex feature-based channel attention UNet network to accurately segment cholangiocarcinoma. The proposed model provides a novel way for layered characteristic extraction and a channel-focusing mechanism, enabling improved high-resolution feature extraction, and enhanced low and high-level feature correlations. Empirical results confirm the superior performance of the model with accuracy, recall, and precision of 0.828, 0.779, and 0.696, respectively. *Rathore et al. (2019)* investigated the automatic precise identification and staging of malignant colorectal cancer specimens. The proposed segmentation method excels in accurately identifying glandular regions in colon tissue and also introduces an innovative algorithm that aims to segment these glandular regions into their component elements. The proposed algorithm performance is similar to that of existing methods for the task of external gland boundary segmentation on the GlaS dataset (Dice Coefficient = 0.87, F-Score = 0.89). *Wang et al. (2021)* targeted the problem of segmentation of colon rectal cancer (CRC) biopsy histopathology data, and they sought to solve the problem of automatic segmentation of CRC intraepithelial neoplasia levels. Using two different preprocessing strategies, pixel-to-propagation consistency (PPC) and Bootstrap Your Own Latent (BYOL), preliminary training of the coder part of the UNet network was performed. The results show that the PPC strategy improves the segmentation performance of UNet more significantly than BYOL. *Shah et al. (2021)* presented an innovative method aimed at improving the accuracy of colorectal cancer segmentation in high-resolution medical images. For this purpose, the developed AtResUnet model, which incorporates atrous convolutions and residual connections in addition to traditional filters achieved high success on the Digest Path 2019 dataset and outperformed existing models in the segmentation of colorectal cancer, especially with a Dice coefficient of 0.748. Furthermore, when used in combination with a simplified approach, a Dice coefficient of 0.753 was achieved.

*Shi et al. (2023)* created the "Enteroscope Biopsy Histopathological Hematoxylin and Eosin Image Dataset for Image Segmentation Tasks (EBHI-Seg)" during their work. This dataset contains a total of 4,456 histopathological images of six distinct stages of tumor differentiation and the anomaly locations of these images are provided as masks. *Shi et al. (2023)* used this dataset for segmentation of histopathological images of colorectal cancer. They used deep learning models namely UNet, SegNet, and MedT as well as machine learning models for segmentation. In this study, each anomaly was segmented separately,

and SegNet was used to obtain the best results for the control group (Dice coefficient = 0.777, Jaccard Index = 0.684) and low-grade intraepithelial neoplasia (Dice coefficient = 0.924, Jaccard Index = 0.864). Similarly, UNet was used for polyp (Dice coefficient = 0.965, Jaccard Index = 0.308), high-grade intraepithelial neoplasia (Dice coefficient = 0.895, Jaccard Index = 0.816), adenocarcinoma (Dice coefficient = 0.887, Jaccard Index = 0.808) and serrated adenoma (Dice coefficient = 0.938, Jaccard Index = 0.886). The disadvantage of the study is that each class is segmented separately. *Tummala et al. (2023)* presented an EffcientNetV2-based classification method for identifying lung and colorectal cancer types in histopathology images. Precise classification of histopathologic images is critical for enhancing the accuracy of diagnosis and quality of treatment. For this purpose, deep learning techniques were used, and a five-class classification task was performed on the LC25000 dataset. The findings indicate that the large model of EfficientNetV2 attains remarkable performance, with an accuracy of 0.999, an AUC (area under the receiver operating characteristic curve) of 0.999, an F1-score of 0.997, and a Matthew correlation coefficient of 0.996. *Mehmood et al. (2022)* proposed a precise and computationally effective method for diagnosing lung and colon cancers. Using a dataset of 25,000 histopathology images, they initially reached an accuracy of 89%. Notably, they employed a novel class selective image processing (CSIP) strategy that enhanced the underperforming class images using contrast enhancement techniques. This approach led to an improved overall accuracy of 98.4%. *Sirinukunwattana et al. (2016)* introduced a neural network that is spatially constrained and tailored to identify cell nuclei in histopathology images related to colon cancer. Their innovative approach involved the utilization of an adjacent group predictor for classifying these nuclei, leading to a remarkable maximum accuracy of 0.971. *Ben Hamida et al. (2021)* have proposed an exploration of deep learning (DL) techniques to examine histopathological images of colorectal cancer, tackling significant challenges within the digital pathology field, such as the scarcity of annotated data and the presence of large image sizes. The study offers a comprehensive evaluation of convolutional neural networks (CNNs), highlighting Resnet's impressive 0.969 accuracy in patch-level classification, and this success extends to public datasets with accuracy rates of 0.997 and 0.998.

# MATERIALS AND METHODS

## General overview

A two-stage solution was developed for the automatic detection of five histopathologic colorectal anomaly types: polyp, low-grade intraepithelial neoplasia, high-grade intraepithelial neoplasia, serrated adenoma and adenocarcinoma, and a control group. Initially, each class was combined and made ready for UNet3+ segmentation. Thus, UNet3+ was able to perform the segmentation process without discriminating the type. UNet3+ was executed with a 10-fold cross-validation technique to generate predicted masks for all samples. After completing the segmentation task, the mask and raw image were multiplied to highlight anomalies in the original images *via* the generated masks. Thus, only the anomaly was left in the image and the relevant part was highlighted. CrossVIT deep learning network was used to determine which class the highlighted part
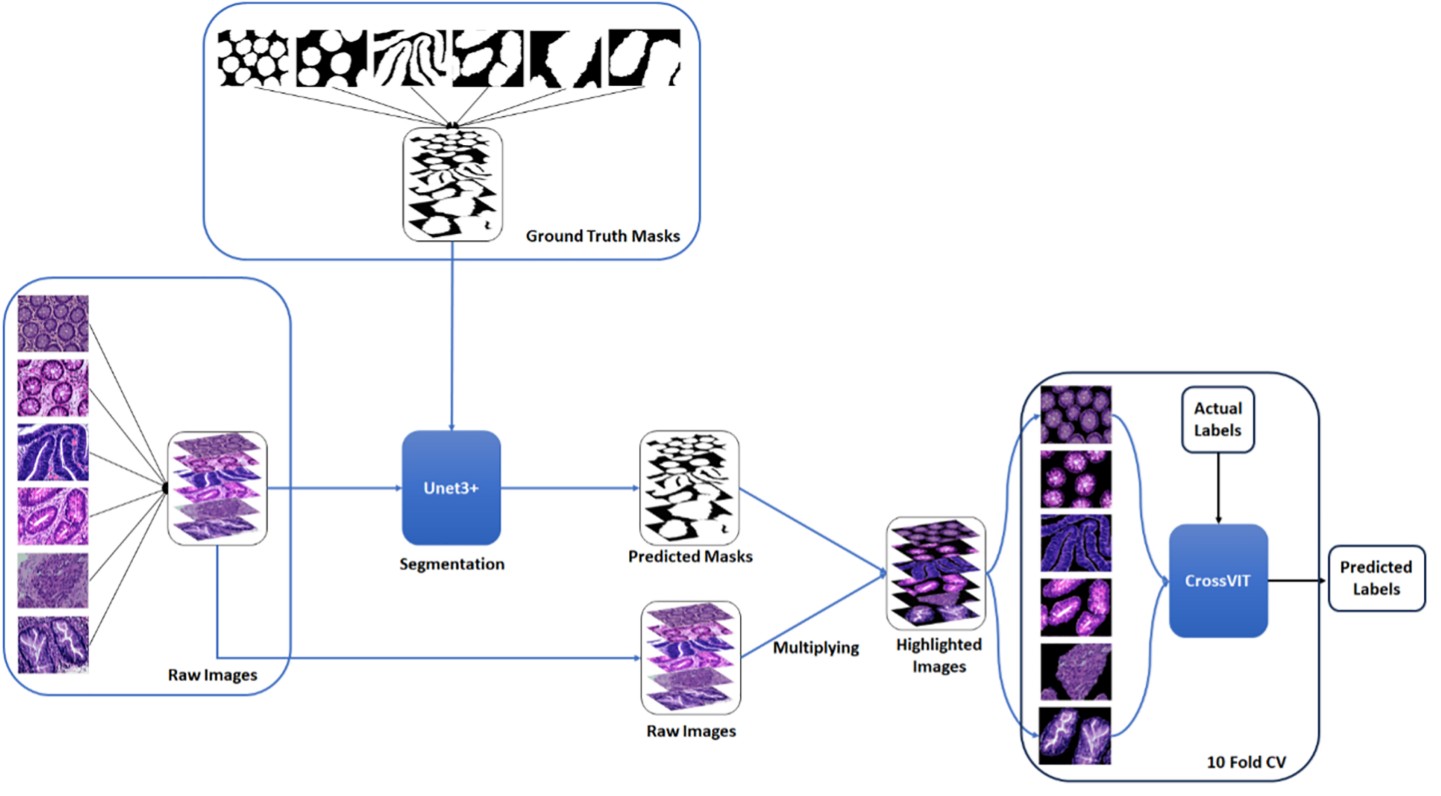

**Figure 1** **An overview of the proposed strategy.** Figure source credit: *MIaMIA Group (2022)*.   

belongs to. In this way, possible anomalies in the sections could be detected with the system created without any external intervention and/or prior knowledge. Figure 1 depicts an overview of the proposed strategy. Figure 2 contains the raw images, the mask obtained after UNet3+ segmentation, and the images obtained by multiplying the raw images and obtained masks, output of proposed method, of each class.

## UNet3+

UNet3+ has become a remarkably effective tool in biomedical image analysis and medical diagnostics. This advanced model extends the basic architecture of UNet, offering higher sensitivity, better resolution, and more capabilities for more complex image segmentation tasks. UNet3+ has already evolved into an essential instrument to support clinical diagnostics processes, especially in biomedical tasks such as the detection, monitoring, and analysis of organs, lesions, or cellular structures (*Huang et al., 2020*).

TheUNet3+ architecture includes many encoder-decoder blocks, which are used to obtain features at different scales starting from the image input. UNet3+ is combined with output layers that are used to combine these features and then determine the boundaries of a particular object or lesion. The model can also be adapted and optimized depending on the desired segmentation task (*Liu, He & Lu, 2022*).

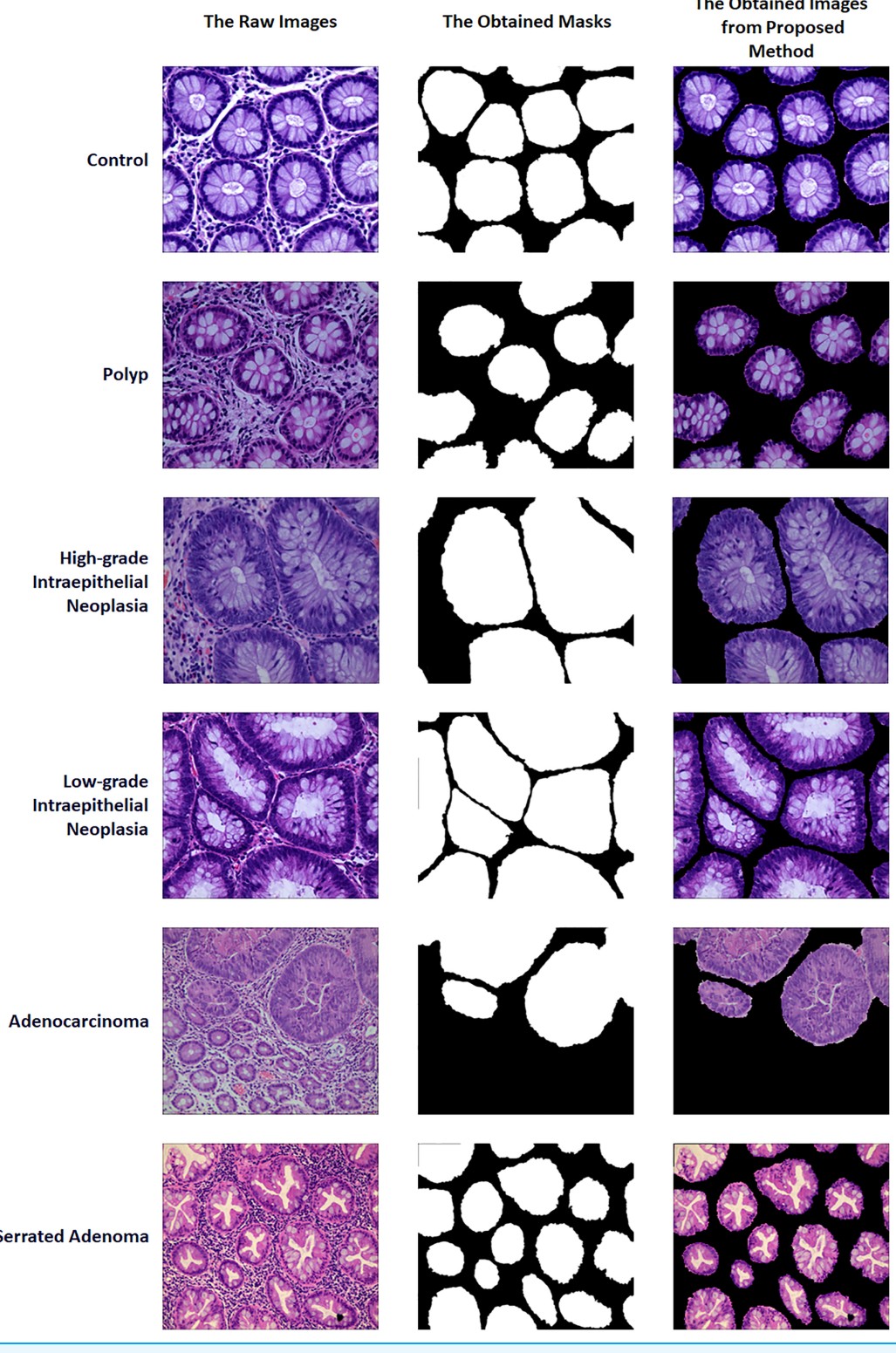

**Figure 2 Sample of raw images, masks and proposed method outputs.** Figure source credit: *MIaMIA Group (2022).*

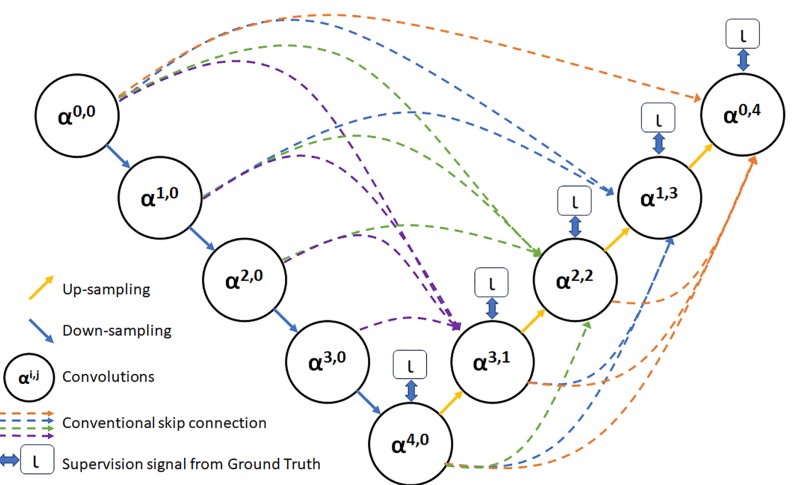

**Figure 3** The UNet3+ network structure.

As depicted in Fig. 3, feature maps $\alpha^{0,0}$, $\alpha^{1,0}$, $\alpha^{2,0}$, $\alpha^{3,0}$, and $\alpha^{4,0}$ are produced following the feature extraction process within each layer. Additionally, feature maps that share a similar size as $\alpha^{3,0}$ are derived *via* distinct pooling transactions. Following this, a convolution operation involving $3 \times 3$ filters with an equivalent number of channels as $\alpha^{3,0}$ is performed to create a feature map with a matching number of channels as $\alpha^{3,0}$. Subsequently, the fusion feature map $\alpha^{3,1}$ is acquired by concatenation and fusion with $\alpha^{3,0}$. Employing a similar approach, fusion feature maps $\alpha^{2,2}$, $\alpha^{1,3}$, and $\alpha^{0,4}$ are produced through identical procedures (*Yousef et al., 2023*).

## CrossVIT

The Cross-Attention Multi-Scale Vision Transformer stands out as a distinctive image transformer designed to obtain multi-scale features for precise classification (*Chen, Fan & Panda, 2021*). This innovative architecture harmonizes images of varying dimensions to improve the quality of visual features for accurate image categorization. It efficiently manages minor and major patch tokens through two distinct branches with differing computing complexities, repeatedly fusing these tokens to reinforce one another. This fusion process is facilitated by a cross-attention module, where each transformer branch serves as a non-patch token agent, enabling information switch through attention mechanisms. This approach not only enables the efficient generation of attention maps in linear time during fusion but also presents a significant improvement over the quadratic time requirements (*Kadri et al., 2022*).

The CrossVIT's architectural framework encompasses a series of K multi-scale transformer encoders. Each of these encoders employs two distinct branches to process tokens from images of several scales ($P_s$ and $P_l$, with $P_s < P_l$) efficiently, culminating in their combination through a module that focuses on cross attention of the CLS tokens. To make the most efficient use of computational resources, our design can flexibly adjust the number of standard transformer encoders in both N and M branches.

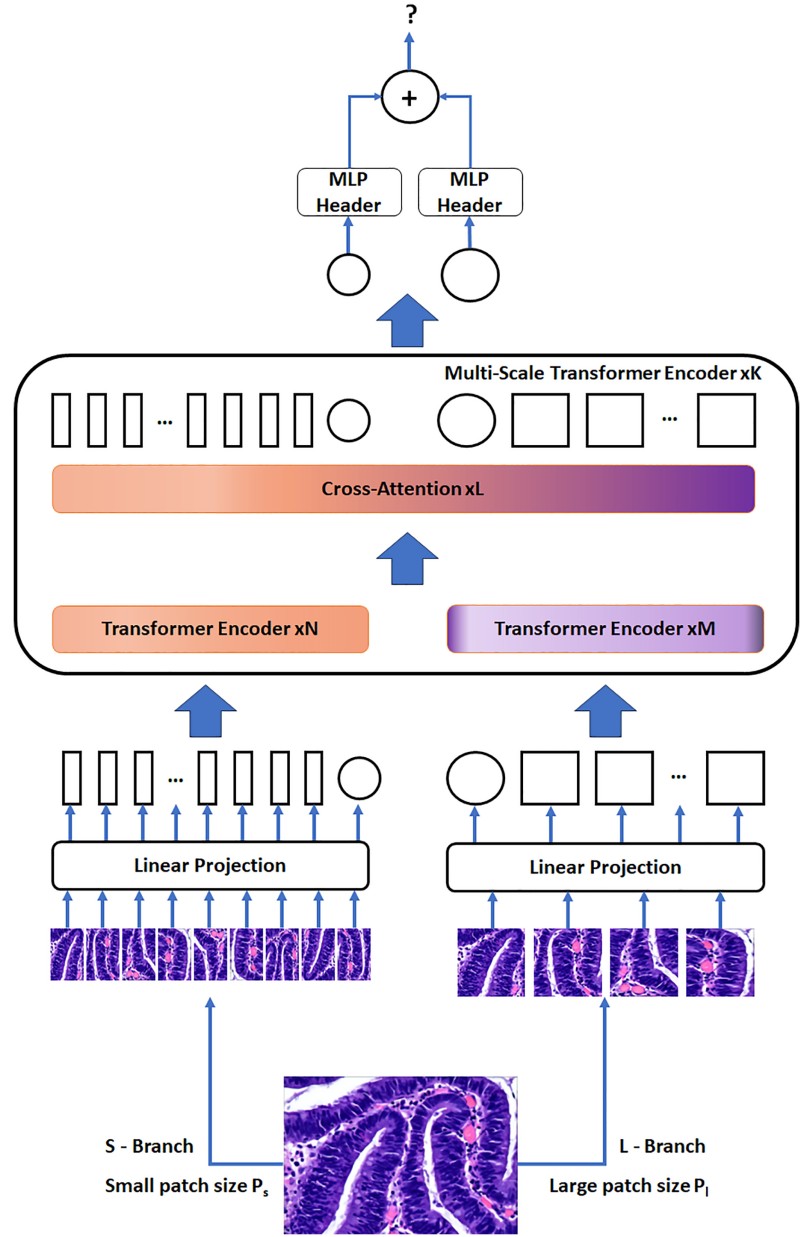

**Figure 4 The network structure of the CrossVIT.** Figure source credit: *MIaMIA Group (2022)*.

Figure 4 visually represents the network structure of the CrossVIT. The CrossVIT predominantly consists of transformer encoders with K scales, each composed of two branches: the L-Branch, the major branch, which employs larger patch sizes ($P_l$), featuring more transformer encoders and broader embedding dimensions, and the S-Branch, the minor branch, which works with smaller patch sizes ($P_s$), encompassing lower embedding dimensions and fewer encoders. These two branches are merged L times, and then the classification token (CLS) tokens at the end of both branches are utilized for predictive tasks. Notably, akin to the Vision Transformer approach (*Sriwastawa & Arul Jothi, 2023*),

CrossVIT incorporates learnable position embeddings for each token within both branches, facilitating the acquisition of position-related information.

## Dataset

The dataset employed in this study was introduced to the literature by *Shi et al. (2023)* and contains a total of 2,226 histopathological images of colorectal cancer anomalies covering five stages of tumor differentiation and a control group called normal: 474 polyps, 637 low-grade intraepithelial neoplasia, 186 high-grade intraepithelial neoplasia, 58 serrated adenomas, and 795 adenocarcinomas and 76 normal. In addition to images of histopathological colorectal cancer slices, the histologically segmented versions of each image and the information about which anomaly was found were determined by clinicians.

Under a light microscope, normal colorectal tissue sections from the control group contain regularly aligned tubular cells, and these cells do not appear infected (*Bilal et al., 2022*). Colorectal polyps resemble normal structures in morphology but have an entirely distinct structural histology. On the other hand, a polyp is an unneeded growth seen on the exterior surface of cells in the body. Polyps are commonly referred to in modern medicine as undesirable enlargements on the mucosa's surface of the tissue (*Ponz de Leon & Di Gregorio, 2001*). The problematic polyp component exhibits a compact luminal structure with minimal cell division. The only difference lies in the slightly higher atomic mass compared to the control group. The most dangerous precancerous lesion is intraepithelial neoplasia (IN). Histological scans demonstrate dense organization, higher branching of adenoid structures, shapes, and various lumen diameters when compared to the normal category. At present, the Padova categorization divides intraepithelial neoplasia into low-grade and high-grade intraepithelial neoplasias. When compared to low-grade IN, high-grade IN exhibits more pronounced structural alterations and nuclear expansion in the lumen. Adenocarcinoma is a type of colon cancer with an uneven distribution of luminal components. During observation, it is difficult to distinguish boundary features, and the nuclei are greatly enlarged (*Ren, 2013*). Serrated adenomas are rarely observed because they constitute less than 1% of all colon polyps. As the endoscopic surface pattern of serrated adenomas is not clearly defined, they may be compared to colonic adenomas with cerebral or tubular crypt openings (*Spring et al., 2006*). The visualization of each class is presented in Fig. 5.

## RESULTS

### Performance evaluation

In this study, all experiments were performed using the k-fold cross-validation method, which entails partitioning the dataset into k equal segments with randomly selected samples. During this procedure, each segment is set aside for testing, while the remaining segments are employed for training. This cycle repeats until each segment has been utilized for testing, ensuring that each sample is employed independently for both testing and training purposes. To obtain an accurate assessment of performance, the models were evaluated using 10-fold cross-validation. A batch size of 32 was used with a learning rate of 0.01 and 50 epochs.

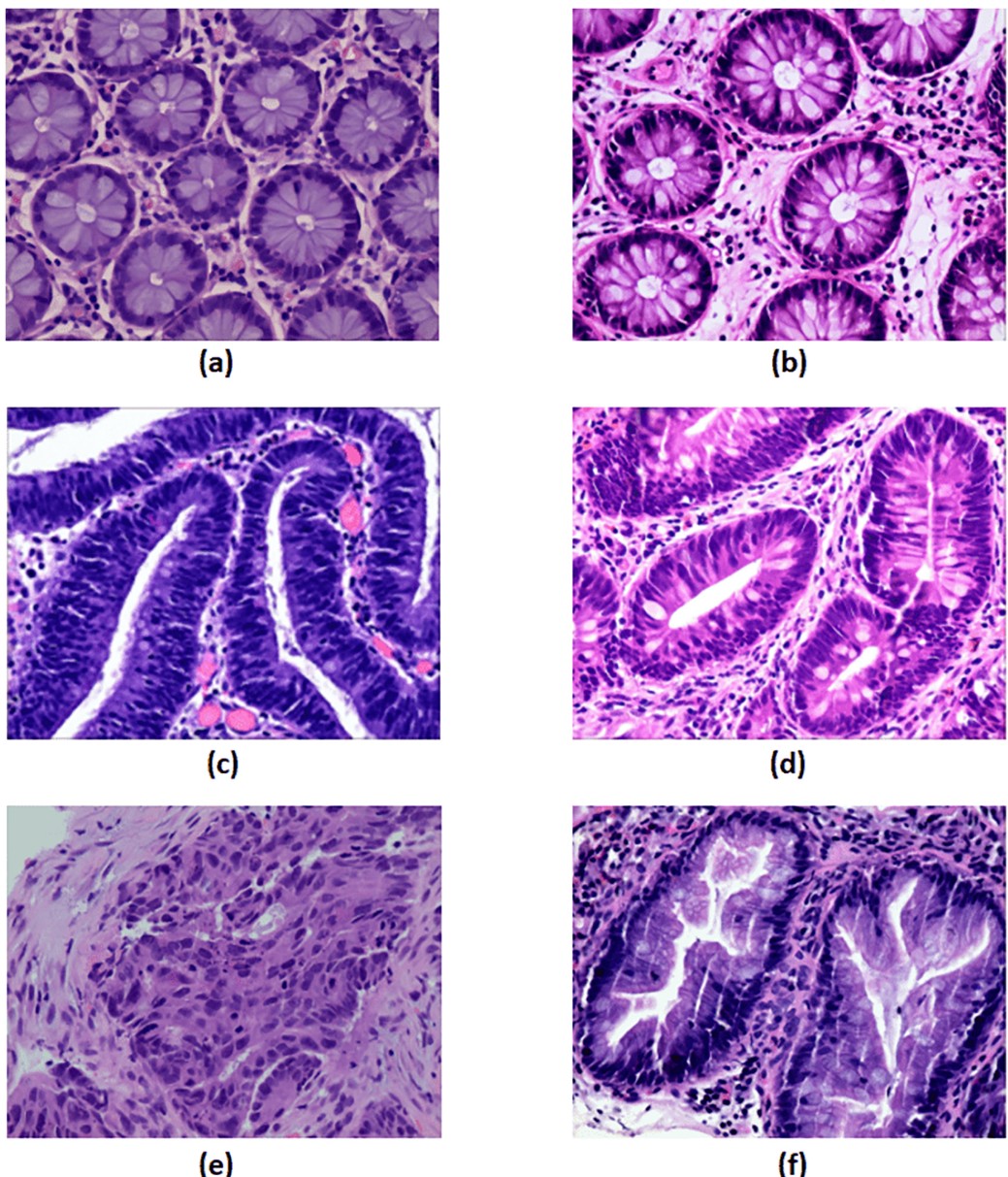

**Figure 5 The visualization of each class.** (A) Control group, (B) polyp, (C) high-grade intraepithelial neoplasia, (D) low-grade intraepithelial neoplasia, (E) adenocarcinoma and (F) serrated adenoma. Figure source credit: *MIaMIA Group (2022)*.

To ensure that the proposed approach is replicable and robust, an evaluation framework has been selected which is rigorous. An extensive 10-fold cross-validation process was applied to all models used, including segmentation and classification. This choice was made to minimize potential biases and to provide a comprehensive evaluation of their performance on various data partitions. During training, a batch size of 32 was used with a learning rate of 0.01, and the training procedure was iterated for a total of 50 epochs. These hyperparameters were carefully adjusted through preliminary experiments to achieve a

balance between model convergence and generalization. All other parameters were kept as default, as in the original models.

Dice coefficient, Jaccard Index, sensitivity, and specificity metrics were used to measure the performance of the segmentation task applied after all histopathology images were combined (*Müller, Soto-Rey & Kramer, 2022*).

The Dice coefficient metric, frequently employed in the assessment of image segmentation algorithms in medical imagery, is a standard measure. It is a validation approach grounded in spatial overlap statistics, gauging the likeness between the algorithm's segmentation results and the ground truth. Equation (1) defines the Dice coefficient.

$$Dice\ coefficient\ =\ 2TP/(2TP\ +\ FP\ +\ FN) \tag{1}$$

The Jaccard index, a well-established measure of set similarity, finds extensive practical use in image segmentation. It quantifies the similarity between a set of samples, representing the ratio of the intersection of the segmentation results and the ground truth to their union. Equation (2) specifies the definition of the Jaccard index. The results fall within the [0,1] range, with higher values indicating superior performance.

$$Jaccard\ Index =\ TP/(TP\ +\ FP\ +\ FN) \tag{2}$$

The sensitivity (Sen), which is also referred to as the true positive rate or Recall, quantifies the proportion of correctly identified positive samples by dividing the number of true positive results by the total number of samples that should have been segmented as positive. Equation (3) provides the precise definition of the Sensitivity.

$$Sen =\ TP/(TP + FN) \tag{3}$$

In segmentation tasks, specificity (Spe) reflects the model's ability to identify. The class representing the background in an image. Given that there is typically a significant number of pixels labeled as background in comparison to the region of interest (ROI), it's common and anticipated to have specificity values close to 1. Therefore, specificity is suitable for verifying the model's capability but may be less indicative of its overall performance. Equation (4) furnishes the exact definition of specificity.

$$Spe =\ TN/(TN + FP) \tag{4}$$

True positives (TP) correspond to pixels correctly identified as belonging to the object. Conversely, true negatives (TN) encompass the pixels accurately categorized as outside the object in both the segmentation and the ground truth. False positives (FP) comprise the pixels erroneously classified by the segmentation as part of the object, even though they do not belong to it. On the other hand, false negatives (FN) include the pixels of the object that the segmentation has incorrectly placed outside.

During the classification stage, the performances of the classification models were evaluated using metrics such as accuracy, F1 score, precision, recall, Matthews correlation coefficient (MCC), and specificity (*Erdaş & Sümer, 2023*).

Accuracy is a metric that measures the proportion of correct predictions made by a model. The accuracy is calculated as the number of correct predictions compared to the total number predicted. The formulation of accuracy is provided by Eq. (5).

$$Accuracy = (TP + TN)/(TP + TN + FP + FN) \tag{5}$$

The F1 score is a measure that combines precision and recall into a single metic, using the harmonic mean. It is widely used in both binary and multi-class classification, providing a more comprehensive assessment of model performance. Equation (6) provides the precise definition of the F1 score.

$$F1\ score = (2TP)/(2TP + FP + FN) \tag{6}$$

Precision is a metric that measures the accuracy of a machine learning model in predicting the positive class. To calculate precision, the number of true positives is divided by the total number of instances predicted as positive (both true and false positives). Equation (7) defines the precision.

$$Precision = (TP)/(TP + FP) \tag{7}$$

Recall is a metric that can be used to measure the frequency with which a machine learning model correctly identifies all true positive examples in a given dataset. To calculate recall, one can divide the number of true positives by the total number of positive examples. Equation (4) furnishes the exact definition of recall.

$$Recall = (TP)/(TP + FN) \tag{8}$$

The Matthews correlation coefficient is a mathematical metric commonly used to evaluate the performance of a classification model, particularly in cases where there is imbalance in the classification problem. Equation (9) specifies the definition of the Matthews correlation Coefficient.

$$MCC = ((TN \times TP) - (FN \times FP)) / \sqrt{(TP + FP)(TP + FN)(TN + FP)(TN + FN)} \tag{9}$$

Specificity is a measure that expresses the proportion of values predicted by a classification model to belong to the negative class over all values that actually belong to the negative class. This measure is similar to recall, but describes the correction in the prediction of negative values. It is also referred to as the true negative rate. Equation (10) provides the precise definition of the specificity.

$$Specificity = (TP)/(TN + FP) \tag{10}$$

For the classification tasks; TP, FP, TN and FN refer to the number of true positives, false positives, true negatives and false negatives, respectively.

**Table 1 Segmentation performance comparison.**

|         | Dice coefficient | Jaccard index | Sensitivity | Specificity |
|---------|------------------|---------------|-------------|-------------|
| UNet3+  | **0.9872**       | **0.9422**    | **0.9832**  | **0.9560**  |
| SegNet  | 0.8978           | 0.8205        | 0.8843      | 0.9076      |
| Unet    | 0.8557           | 0.8469        | 0.8571      | 0.8568      |
| MedT    | 0.7821           | 0.6647        | 0.8011      | 0.7986      |

**Note:**
  Bold shows the best results.

## Empirical results and findings

The subject matter explored in this research, which pertains to anomaly detection and anomaly type classification, can be theoretically framed within the domain of artificial intelligence as a combined task involving image segmentation and classification. The outcomes of the experiments designed for anomaly segmentation are presented in Table 1. In this context, classification experiments were carried out for different algorithms using the outputs of the model with the best segmentation result. Table 2 presents the results of the experiments conducted to classify anomaly types and validate the efficacy of the proposed methodology.

Upon analyzing the segmentation results in Table 1, it can be observed that UNet3+ outperforms the other methods particularly in this regard. Specifically, the Dice coefficient is as high as 0.9872, indicating a strong overlap between segmentation and ground truth. The Jaccard Index of 0.9422 further confirms significant set similarity. The sensitivity metric demonstrates the model's ability to accurately identify positive examples with a value of 0.9832. Additionally, the specificity metric indicates the model's ability to recognize the background class, achieving a score of 0.9560.

According to the results, SegNet surpasses Unet and MedT, although it lags behind UNet3+ with a Dice coefficient of 0.8978, Jaccard Index of 0.8205, sensitivity metric of 0.8843, and specificity value of 0.9076. Unet appears to have slightly lower scores than the SegNet, with a Dice coefficient of 0.8557, Jaccard Index of 0.8469, sensitivity metric of 0.8571, and specificity value of 0.8568. When the MedT method is analyzed, the results obtained are as follows: Dice coefficient of 0.7821, Jaccard Index of 0.6647, sensitivity Metric of 0.8011, and specificity value of 0.7986. When the results obtained are considered, MedT has the worst performance among the methods tested.

The study's classification findings were sorted into three distinct groups for a thorough analysis: evaluating models trained on raw histopathological images, obtained masks derived from segmentation, and the proposed method. The study proceeded with classification experiments utilizing Unet3+, since it was observed that Unet3+ proved to be effective in the segmentation experiments. This methical approach enabled a comprehensive assessment of the effectiveness and performance of each technique. Table 2 presents the classification results obtained using various methods, including the proposed CrossVIT method, ResNetV2, InceptionNetV3, and EfficientNetV2. According to the results, it appears that the CrossVIT method performed best when fed with the multiplication of masks derived from segmentation and raw images. This approach has

**Table 2 Results of Classification Models.**

| | | Accuracy | F1-score | Precision | Recall | MCC | Specificity |
|---|---|---|---|---|---|---|---|
| CrossVIT | The raw images | 0.8545 | 0.8562 | 0.8462 | 0.8984 | 0.8562 | 0.8960 |
| | The obtained masks | 0.8176 | 0.7482 | 0.8660 | 0.6934 | 0.7538 | 0.8929 |
| | **The proposed method** | **0.9340** | **0.9037** | **0.9446** | **0.8723** | **0.9102** | **0.9849** |
| ResNetV2 | The raw images | 0.8032 | 0.6821 | 0.7517 | 0.6662 | 0.6956 | 0.8284 |
| | The obtained masks | 0.7448 | 0.6787 | 0.6887 | 0.6536 | 0.6254 | 0.7198 |
| | The proposed method | 0.8747 | 0.8051 | 0.8223 | 0.8349 | 0.7988 | 0.8374 |
| InceptionNetV3 | The raw images | 0.8298 | 0.8522 | 0.8569 | 0.8475 | 0.6959 | 0.8066 |
| | The obtained masks | 0.7648 | 0.7483 | 0.7051 | 0.7262 | 0.6248 | 0.7455 |
| | The proposed method | 0.8651 | 0.9033 | 0.9033 | 0.8543 | 0.7597 | 0.8467 |
| EfficientNetV2 | The raw images | 0.8318 | 0.8664 | 0.8590 | 0.8427 | 0.6834 | 0.8059 |
| | The obtained masks | 0.7933 | 0.8192 | 0.8038 | 0.8353 | 0.6482 | 0.7476 |
| | The proposed method | 0.9031 | 0.9007 | 0.9216 | 0.8720 | 0.8232 | 0.8644 |

**Note:**
Bold shows the best results.

produced the most favorable outcomes, with values of 0.9340 for Accuracy, 0.9037 for F1 score, 0.9446 for precision, 0.8723 for recall, 0.9102 for MCC, and 0.9849 for Specificity metrics, respectively. Irrespective of the preprocessing method employed, it can be observed that the CrossVIT method outperformed all other methods. From the raw images, results of 0.8545, 0.8562, 0.8462, 0.8984, 0.8562, and 0.8960 were obtained for accuracy, F1 score, precision, recall, MCC, and specificity metrics, respectively. In contrast, when the mask obtained after segmentation was used in the experiment, the corresponding measurements achieved values of 0.8176, 0.7482, 0.8660, 0.6934, 0.7538, and 0.8929 for the mentioned metrics, respectively.

It can be seen that EfficientNetV2, although the results obtained on the raw histopathological images, on the masks obtained from the segmentation and on the images generated by the proposed method were promising, could not surpass the classification performance of CrossVIT. It is worth noting that both ResNetV2 and InceptionNetV3 models achieved comparable results, with InceptionNetV3 showing slightly better classification performance. The proposed method appears to yield the most favorable classification results, irrespective of the classification models employed. In contrast, the results obtained with raw histopathological images are inferior to those obtained with the proposed method, but superior to the performance obtained with masks derived from segmentation.

## DISCUSSION AND CONCLUSIONS

This study addressed the challenges associated with colorectal cancer diagnosis, which is a significant global health concern. Accordingly, this research emphasizes the requirement for improved diagnostic tools within this domain. The current diagnostic methods have limitations such as subjectivity, data overload, and difficulties in processing big data, necessitating the development of more efficient and objective diagnostic tools. The

proposed methodology adopts a two-stage approach consisting of image segmentation and anomaly classification, to enhance the accuracy of diagnosing colorectal cancer.

When analyzing the segmentation obtained using the UNet3+ model, the results display a high degree of promise in terms of Dice coefficient, Jaccard Index, sensitivity, and specificity. The results obtained indicate that the proposed model effectively recognizes anomalies in histopathological images, with a strong overlap with the ground truth and the ability to accurately identify positive samples and the background class. When the segmentation results obtained are compared with the findings of other segmentation methods, the success of the UNet3+ method used in the study is revealed in all the Dice coefficient, Jaccard Index, sensitivity metrics. This situation can be explained by the fact that the UNet and SegNet and MedT models lagged behind UNet3+ in terms of architecture.

In evaluating anomaly classification, models that were trained on raw images, models that solely utilized segmentation-based masks, and the proposed approach were compared.

The results of the classification, obtained using various methods, including the proposed CrossVIT method, ResNetV2, InceptionNetV3, and EfficientNetV2, indicate that the CrossVIT method performed best when fed with the multiplication of masks derived from segmentation and raw images. The superior performance of the suggested method can be attributed to its elimination of the extraneous components surrounding the relevant anomaly; this way, the CrossVIT deep learning model can solely concentrate on the anomaly. The lower results achieved through using masks formed from segmentation in comparison to raw images are due to the emphasis only on the anomaly's shape in the formed masks, with no accompanying information on its contents. By its nature, CrossVIT has an architecture that can detect smaller details and general patterns from two separate branches. When this is combined with the proposed method, CrossVIT can obtain information about both the general structure and shape of the anomaly and the small patterns in the internal structure of the relevant anomaly. Regardless of the classification models used, the proposed preprocessing method appears to consistently produce the most favorable classification results. It is noteworthy that although the performance obtained on raw histopathological images exceeds that obtained with masks derived from segmentation, it doesn't reach the effectiveness demonstrated by the proposed method.

This study has several limitations. Although the dataset used in the study includes different types of colorectal anomalies, the results should be evaluated on different datasets to strengthen the effectiveness of the proposed methods. Furthermore, all methods are in their vanilla form. Classification success can be further improved by integrating different feature selection methods into the system. It is believed that the UNet3+ segmentation method has reached saturation for this dataset, but the segmentation performance of other methods can be improved using techniques such as PPC and BYOL. Future research can focus on further enhancing the proposed methodology, expanding the dataset to improve generalization, and investigating the integration of multi-modal data for comprehensive cancer diagnosis. Additionally, efforts can be directed toward addressing the challenges of computational efficiency to enable real-time clinical applications.

### Funding
The authors received no funding for this work.

### Competing Interests
The authors declare that they have no competing interests.

### Author Contributions
- Çağatay Berke Erdaş conceived and designed the experiments, performed the experiments, analyzed the data, performed the computation work, prepared figures and/or tables, authored or reviewed drafts of the article, and approved the final draft.

### Data Availability
The data is available at figshare: MIaMIA Group (2022). EBHI-SEG. figshare. Dataset. https://doi.org/10.6084/m9.figshare.21540159.v1.

### Supplemental Information
Supplemental information for this article can be found online at http://dx.doi.org/10.7717/peerj-cs.2071#supplemental-information.

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
