# Peer review of "Computer-aided colorectal cancer diagnosis: AI-driven image segmentation and classification"

_PeerJ Computer Science, doi:10.7717/peerj-cs.2071_

## Round 0.1 · original submission · Major Revisions

The reviewers have substantial concerns about this manuscript. The authors should provide point-to-point responses to address all the concerns and provide a revised manuscript with the revised parts being marked in different color.

Reviewer 1 ·

Basic reporting

This paper narrates a comprehensive account of imaging diagnostics for Colorectal Cancer. The language used is professional, and the references are well-cited. The logical flow is strong, with robust rationale discussions and solid results. However, please enhance the resolution of your figures.

Experimental design

This paper outlines the method step by step, with clear logic. The experimental design is well-suited for publication.

Validity of the findings

This paper describes computer-aided methods to enhance the accuracy of cancer diagnostics. The data and method details are provided. Conclusions are well stated.

Additional comments

Kindly elaborate on the distinctions between your paper titled "Computer-Aided Colorectal Cancer Diagnosis: AI-Driven Image Segmentation and Classification" and "Artificial Intelligence-Aided Endoscopy and Colorectal Cancer Screening." (https://www.mdpi.com/2075-4418/13/6/1102)

Reviewer 3 ·

Basic reporting

The manuscript titled "Computer-aided colorectal cancer diagnosis: AI-driven image segmentation and classification" presents a novel two-stage diagnostic tool for cancer using deep learning methods. The methods using UNET3+ for image segmentation and then using CrossVIT for cancer type classification. By applying on the colorectal cancer dataset, the proposed method demonstrate a good performance through various metrics. The topic of the paper is significant, however, I have a few concerns that would like the authors to address:
1. Unclear description of rationale: In line 50, " Nevertheless, the following issues are often encountered during the process of diagnosis: The microscope can be set to high magnification when fine structures need to be observed in more detail". What's the "encountered issue"? In line 62, "With the developing technology, the type of segmented object can also be found by using segmentation but this is not valid for histopathological images because they contain only one type of object. In other words, pathologists can only apply segmentation for the suspected type of anomaly". It's unclear about the meaning of "object", "type", and the reason of "not valid".

Experimental design

Although the author wrote a summary of related study in the manuscript, a more comprehensive literature review is definitely needed. For example, in line 82, the author stated that "as far as known, no study has used both methods simultaneously". However, I found at least three paper doing both segmentation and classification on cancer data, even on the same type of cancer (colorectal cancer):
1. Bokhorst, JM., Nagtegaal, I.D., Fraggetta, F. et al. Deep learning for multi-class semantic segmentation enables colorectal cancer detection and classification in digital pathology images. Sci Rep 13, 8398 (2023).
2. Ramesh, S., Sasikala, S., Gomathi, S. et al. Segmentation and classification of breast cancer using novel deep learning architecture. Neural Comput & Applic 34, 16533–16545 (2022).
3. Zhou, Yue, et al. "Multi-task learning for segmentation and classification of tumors in 3D automated breast ultrasound images." Medical Image Analysis 70 (2021): 101918.

Please also read this and compare the proposed method with the existing methods.

Also, when describing the methods, some terminologies need to be explained advance, such as X in figure 2, Ps, Pl, CLS token, MLP. Also, the numbers should be superscript for the convolution X. The performance metrics of classification also need a detailed explanation.

Validity of the findings

Although the idea of proposed methods is introduced, more details are needed for reproducibility, such as the choice of 10-fold, value of hyper-parameters, model architecture specifics, training procedures. Those require more elaboration. Moreover, the comparison between the proposed method and other segmentation methods is not fair. Instead of directly use the numbers in previous study, it's better to follow their procedure but rerun it to eliminate technical factors. The specificity should also be evaluated for other methods. Table 1 and 2 are redundant. Also, numerous classification methods exist but the author didn't compare CrossVIT with other methods. Moreover, the author should also compare the overall pipeline with other published segmentation+classification pipeline. It will be better if the methods can also be compared on other colorectal cancer datasets or even other type of cancer.

Reviewer 4 ·

Basic reporting

The authors explored a novel method that combined both image segmentation and classification method to diagnose the colorectal cancer in the manuscript.In general, the manuscript was written in a relatively clear English but there are some major issues as follows:
1). In the background/context section, authors need to provide visualization of each colorectal cancer type up-front with detailed legends, instead of demonstrating them later in the dataset section, to help readers understand the background more easily.
2). Additionally, Fig 4 alone is lack of detailed legends, the author put too much descriptive texts in the Dataset section, instead of commenting directly in the legend of Fig 4, which can help readers understand each colorectal cancer class more easily.
3). Moreover, authors need to add a few more real image examples of raw images after Unet3+ processing, and after Masks multiplying with raw images, to demonstrate the uniqueness of author's approach (masks multiplying the raw images). Since multiplication of masks after Unet3+ with raw images is unique here in the manuscript, the authors need to further expand details on this multiplication step.
4). In the Related Studies section, the author provided too much descriptive texts. The author should try to come up with a summary table clearly lay out each benchmark method's data attributes, model structure and corresponding performance metrics.

Experimental design

The approach proposed by the author is mainly composed of 2 major parts: 1). Unet3+ mask generation and 2). CrossVIT classification. The manuscript could benefit more if:
1) the author can further expand Unet3+ applications, showing real mask generation examples in addition to the Unet3+ model structures in the Fig 2. Additionally, Fig 2 is directly taken from the literature, failed to demonstrate author's work here in the manuscript. And Table 1 is redundant here, it is just the last row of Table 2.
2). the author mentioned the multiplication of masks generated from Unet3+ with raw images, however, such step is missing details and image demonstration in the whole manuscript. The author need to further expand on the multiplication step.
3). In the crossVIT section, the data preprocessing step is missing. The author only showed the general structure of the crossVIT network, without detailed legend comments to help readers better understand the function of crossVIT and potential replicate such part. For Table 3, comparison against other benchmark methods need to added for the final diagnosis evaluation.

Validity of the findings

There are several major concerns on the findings and results reporting of the manuscript.
1). In the results section, line 302, 310, 317, the authors need to give more explanation on the definition of TP, FP, FN here, since TP, FP, TN, FN definitions are different in the image segmentation task and image classification task, which are both explored in the manuscript.
2). In the abstract section, line 29 to 30, the value of metrics does not match with metric names. Specifically, 4 metric values are provided for 6 metric types.
3). Line 92, full name of AJI score should be given up-front. Similarly, for line 223, the full name of crossVIT should be given at its first appearance in the abstract section.
4). Table 1 is redundant here, it is just the last row of Table 2. For Table 3, comparison against other benchmark methods need to added for the final diagnosis evaluation.
5). In all evaluation tables, the best results group should be highlighted in bold/italic. This is a common practice in the academia paper publish.

Reviewer 5 ·

Basic reporting

The study of "Computer-aided colorectal cancer diagnosis: AI-driven image segmentation and classification" used a two-step model (segmentation and five-class classification) for diagnosing colorectal cancer. However, there are some significant concerns need to be revised and improved.

Experimental design

Experiments are not well designed to achieve the goal of the study.
For databases, why do you only use EBHI-Seg from Shi et al. (2023)? You also mentioned another five-class classification research using LC25000 dataset so you can make some comparisons with them.
For preprocessing, PPC and BYOL were mentioned in the “Related Studies” and results showed PPC can improve the segmentation performance of UNet. Why don’t you use PPC for your study?
For methodologies:
As mentioned in Hamida et al. (2021), SegNet is better than UNet. And in the study of Shi et al. (2023), UNet and SegNet were used for different categories. But you used UNet for segmentation of all data without further exploration of two methods. This is not reasonable.
Same for classification part, resnet was mentioned in “Related Studies”. Why do you only use CrossVIT? More explanations are needed.
Also, a feature selection method GWO-RF mentioned in “Related Studies” shows a better sensitivity 0.986. Do you have any explorations about this for your study?

Validity of the findings

Results of this study are unconvincing because of the bad experimental design, and there’re some vague or wrong expressions.
In “Abstract”, “a classification performance of 0.9872, 0.9422, 0.9832, and 0.9560 for Accuracy, F1 score, Precision, Recall, Matthews Correlation Coefficient, and Specificity, respectively” is lack of corresponding performance metrics; “by achieving high sensitivity in both the identification of anomalies and the segmentation of regions” is wrong because no sensitivity data shown for classification part.
In “Related Studies”, line 82, “As far as known, no study has used both methods simultaneously” needs to be revised. There’re many studies combining image segmentation and classification. Even the related studies you mentioned from Hamida et al. (2021) used segmentation and classification.
Line 178-180, “Furthermore, the research delves into pixel-wise segmentation using SegNet and UNet models, employing a multi-stage training approach to enhance accuracy, resulting in rates reaching up to 0.812 and 0.991” can’t be concluded from Hamida et al (2021) based on “UNET and SEGNET are used and tested in different training scenarios including data augmentation and transfer learning and ensure up to 76.18% and 81.22% accuracy rates. Besides, we test our training strategy and models on the CRC-5000, NCT-CRC-HE-100K and WARWICK datasets. Respective accuracy rates of 98.66%, 99.12% and 78.39% were achieved by SEGNET.” and “SEGNET enables ≈5% higher accuracy rate than UNET” from their article.

Additional comments

Major contributions of the study can be listed at the end of the introduction to show the importance of your study.
No limitations mentioned for your study in “Discussion & Conclusions”.
Figure 4, “(a) control group, (b) polyp, (c) high-grade intraepithelial neoplasia, (d) low-grade intraepithelial neoplasia, (e) adenocarcinoma and (f) serrated adenoma” should be added for the self-explanatory.

---

## Round 0.2 · Minor Revisions

The reviewers has a minor concerns. Authors should carefully address the concerns.

Reviewer 1 ·

Basic reporting

no comment

Experimental design

no comment

Validity of the findings

no comment

Additional comments

no comment

Reviewer 5 ·

Basic reporting

All my concerns have been well addressed except GWO-RF related comment:
"Also, a feature selection method GWO-RF mentioned in “Related Studies” shows a better sensitivity 0.986. Do you have any explorations about this for your study?"

If you think "it distorts the meaning and integrity of the study", this shouldn't be the related work, and it may need to be removed.

Other than that, the manuscript is ready for publication.

Experimental design

The study has detailed and reasonable experimental design.

Validity of the findings

Results are good and clear.

---

## Round 0.3 · accepted · Accept

The remaining minor concerns have been addressed.